# New approach for membrane protein reconstitution into peptidiscs and basis for their adaptability to different proteins

**Gabriella Angiulli[1†], Harveer Singh Dhupar[2†], Hiroshi Suzuki[1], Irvinder Singh Wason[2], Franck Duong Van Hoa[2]\*, Thomas Walz[1]\***

[1]Laboratory of Molecular Electron Microscopy, Rockefeller University, New York, United States; [2]Department of Biochemistry and Molecular Biology, University of British Columbia, Vancouver, Canada

**Abstract** Previously we introduced peptidiscs as an alternative to detergents to stabilize membrane proteins in solution (Carlson et al., 2018). Here, we present 'on-gradient' reconstitution, a new gentle approach for the reconstitution of labile membrane-protein complexes, and used it to reconstitute *Rhodobacter sphaeroides* reaction center complexes, demonstrating that peptidiscs can adapt to transmembrane domains of very different sizes and shapes. Using the conventional 'on-bead' approach, we reconstituted *Escherichia coli* proteins MsbA and MscS and find that peptidiscs stabilize them in their native conformation and allow for high-resolution structure determination by cryo-electron microscopy. The structures reveal that peptidisc peptides can arrange around transmembrane proteins differently, thus revealing the structural basis for why peptidiscs can stabilize such a large variety of membrane proteins. Together, our results establish the gentle and easy-to-use peptidiscs as a potentially universal alternative to detergents as a means to stabilize membrane proteins in solution for structural and functional studies.

**\*For correspondence:**
fduong@mail.ubc.ca (FDVH);
twalz@rockefeller.edu (TW)

[†]These authors contributed equally to this work

## Introduction

Integral membrane proteins make up a third of the human proteome and are the target of many therapeutic drugs, but until recently, determining their structure has been very challenging. Recent advances in single-particle cryo-electron microscopy (cryo-EM) now allow structure determination without the need to grow high-quality crystals, one of the major bottlenecks for structure determination by X-ray crystallography. Detergents are the most commonly used way to stabilize membrane proteins in solution outside a lipid bilayer, but detergents are cause for concern. Because of their amphipathic nature, small size and dynamic micellar state, detergents not only cover the hydrophobic belt of a membrane protein, but may also destabilize their fold by disrupting hydrophobic interactions. This structural destabilization may only slightly affect the conformation of proteins, but it can also cause various degrees of denaturation, which over time will result in the aggregation and precipitation of the proteins. Substantial efforts have thus been devoted to finding alternative ways to stabilize membrane proteins in solution, especially for structural studies, which resulted in the introduction of several novel membrane mimetics, such as amphipols (*Diab et al., 2007*; *Tribet et al., 1998*), membrane-scaffold protein (MSP)-based lipid nanodiscs (*Civjan et al., 2003*; *Denisov and Sligar, 2017*), saposin A-based Salipro particles (*Frauenfeld et al., 2016*), and styrene maleic acid co-polymer lipid particles (SMALPs) (*Bada Juarez et al., 2019*; *Morrison et al., 2016*), all reviewed in *Autzen et al. (2019)*.

Recently, our laboratory introduced the peptidisc as a novel type of peptide scaffold designed to keep membrane proteins stable in solution without detergent (*Carlson et al., 2018*). The peptidisc peptide is based on the original apolipoprotein A-I peptide initially reported by the Segrest group

(*Chung et al., 1985*), but has the reversed amino-acid sequence. This peptide forms two amphipathic helical stretches that are separated by a proline residue. When the peptidisc peptide is mixed with detergent-solubilized membrane protein and the detergent is removed, multiple copies of the peptide associate with the transmembrane domain, embedding the protein in a peptidisc, which can also incorporate lipids that may be bound to the membrane proteins (*Carlson et al., 2019*; *Carlson et al., 2018*). We have shown that the peptidisc peptide works equally well for the reconstitution of α-helical and β-barrel membrane proteins and that reconstitution is simple, efficient and requires little optimization. Because of the facile auto-assembly process, peptidisc formation is possible through rapid detergent removal in the presence of the peptidisc peptide using 'on-column' and 'on-bead' methods (*Carlson et al., 2018*).

Here, we present a new gentle method to reconstitute membrane proteins into peptidiscs, which we named 'on-gradient' reconstitution and we believe should be particularly useful for labile membrane-protein complexes. As model system, we used the reaction center (RC) of *Rhodobacter sphaeroides* that can be purified on its own, or surrounded by a light-harvesting complex 1 (LH1) ring, or as a large dimeric complex consisting of two RC–LH1 complexes linked together by PufX proteins (RC–LH1–PufX complex). All three RC complexes could be stabilized by peptidiscs, showing that peptidiscs can be used for membrane-protein complexes of diverse sizes. The RC–LH1–PufX complex is particularly demanding for the membrane mimetic because of its large, convoluted and bent transmembrane domain (*Qian et al., 2013*). In addition, we show that peptidisc-stabilized membrane proteins (in this case reconstituted with the conventional 'on-beads' method) can be used for high-resolution structure determination by single-particle cryo-EM. Using *Escherichia coli* MsbA, we show that peptidiscs stabilize this ABC transporter as effectively as MSP-based lipid nanodiscs (*Mi et al., 2017*). In particular, the cryo-EM map at 4.2 Å resolution reveals that the peptidisc preserves the native conformation of MsbA as well as its interaction with its lipopolysaccharide (LPS) cargo. We also present a 3.3 Å resolution cryo-EM structure of the homo-heptameric mechanosensitive channel MscS from *E. coli*. Notably, the arrangements of the peptidisc peptides around MsbA and MscS are very different, revealing the structural basis for how the peptidisc scaffold can adapt to membrane proteins of different sizes, shapes and symmetries.

Together, our results demonstrate that peptidiscs can be used to stabilize membrane proteins and membrane-protein complexes with very different structures, that they preserve the native conformation of membrane proteins as well as their interaction with associated lipids, and that they allow for high-resolution structure determination by single-particle cryo-EM.

## Results

### On-gradient reconstitution for the stabilization of membrane-protein complexes

We developed a new, gentler method to reconstitute membrane proteins into peptidiscs. In this approach, the detergent-solubilized membrane protein is mixed with an excess of peptidisc peptide, and the mixture is overlaid on a detergent-free linear sucrose gradient. As the gradient is centrifuged, proteins enter the detergent-free gradient and become reconstituted into peptidiscs, whereas excess peptide and detergent micelles remain in the overlaid solution (*Figure 1A*). In addition, since the sucrose gradient separates proteins according to their density, contaminants and aggregates will migrate to different positions in the gradient. To test whether this approach allows efficient reconstitution and effective separation, we used the *R. sphaeroides* RC core complex by itself (99 kDa), the monomeric RC–LH1 complex (258 kDa), and the dimeric RC–LH1–PufX complex (521 kDa). In addition to their varied sizes, these complexes are colored due to their pigmented cofactors, making it easy to follow their migration into the sucrose gradient upon centrifugation.

After on-gradient reconstitution into peptidiscs, the three complexes formed clearly visible bands at distinct positions in the sucrose gradient (*Figure 1B*). These bands were located in the same positions as the detergent-solubilized complexes run on detergent-containing sucrose gradients (*Figure 1B*), showing that the peptidisc does not affect the overall hydrodynamic properties of the complexes. Furthermore, when these complexes were recovered from the sucrose gradients,

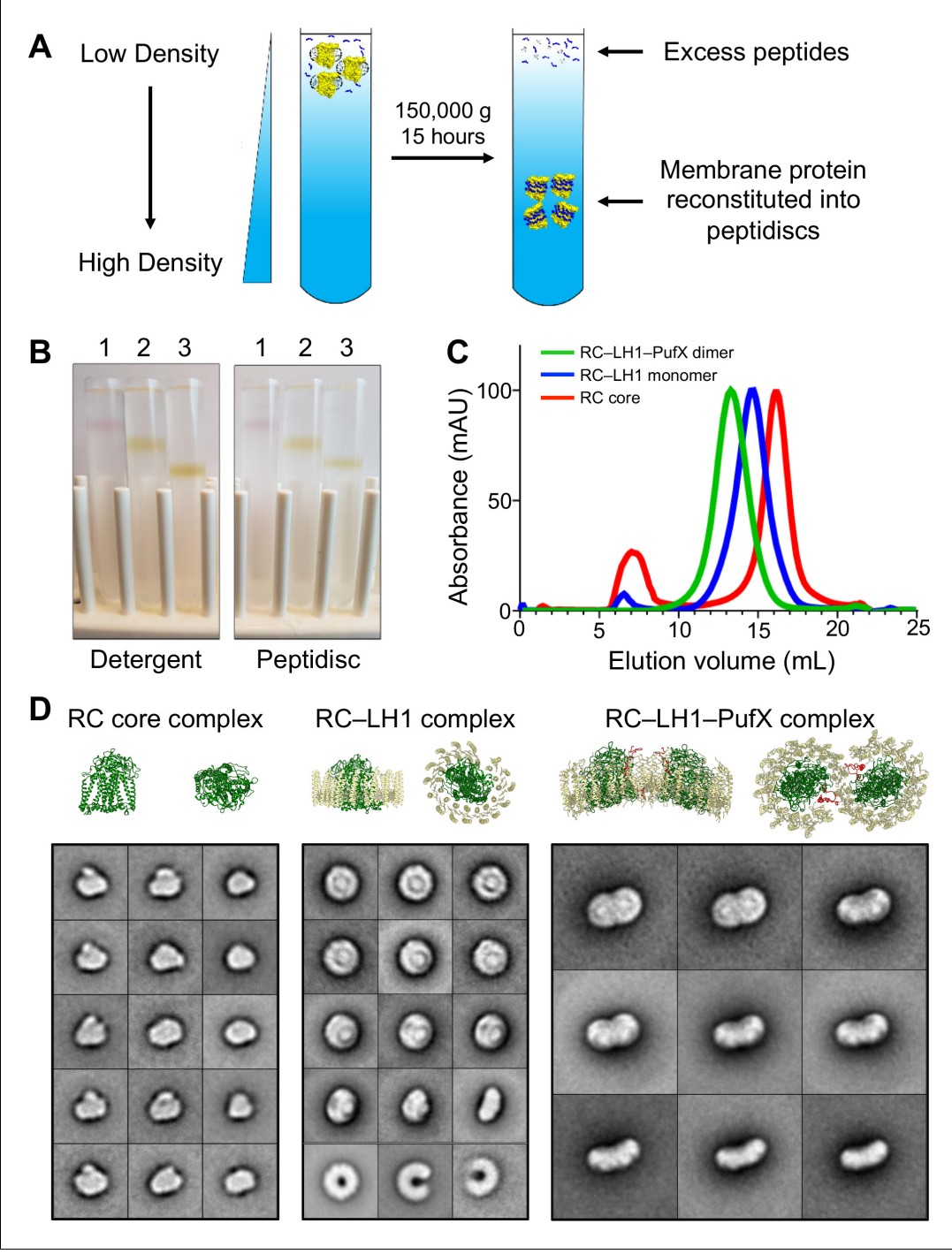

**Figure 1.** Reconstitution of *Rhodobacter spheroides* reaction center (RC) complexes into peptidiscs by using the 'on-gradient' method. (**A**) Schematic drawing showing the principle of the on-gradient reconstitution method: detergent-solubilized membrane protein is mixed with an excess of peptidisc peptide and the mixture is overlaid onto a detergent-free linear sucrose gradient. Upon centrifugation, the protein reconstitutes into peptidiscs and localizes to a discrete band in the gradient, whereas excess peptides and detergent micelles stay at the top. (**B**) Gradients of the colored *R. sphaeroides* RC complexes showing their migration in the presence of detergent (left panel) and after reconstitution into peptidiscs (right panel). 1: RC core complex (99 kDa), 2: monomeric RC–LH1 complex (258 kDa), and 3: dimeric RC–LH1–PufX complex (521 kDa). (**C**) Size-exclusion chromatography profiles of RC complexes reconstituted into peptidiscs. Traces are normalized to 100 mAU. (**D**) Selected 2D-class averages of the three RC complexes (see *Figure 1—figure supplement 1* for all 100 class averages). For comparison, ribbon representations are shown for the *R. sphaeroides* RC (PDB: 1K6L; *Pokkuluri et al., 2002*), the *Rhodopseudomonas*

*Figure 1 continued*

*palustris* RC–LH1 complex (PDB: 1PYH; *Roszak et al., 2003*) and the *R. sphaeroides* RC–LH1–PufX complex (PDB: 4V9G; *Qian et al., 2013*). RC: green, LH1: yellow, and PufX: red. The bottom row of the class averages for the RC–LH1 complex show empty LH1 rings that were present in the preparation. The side lengths of the individual class averages are 24.4 nm for the RC core complex, 32.5 nm for the monomeric RC–LH1 complex, and 48.8 nm for the dimeric RC–LH1–PufX complex.

The online version of this article includes the following figure supplement(s) for figure 1:

**Figure supplement 1.** Negative-stain electron microscopy analysis of RC complexes.

**Figure supplement 2.** Reconstitution of MsbA with fluorescent peptidisc peptide using the 'on-gradient' method.

concentrated and analyzed by size-exclusion chromatography (SEC), all samples produced a sharp, symmetric peak (*Figure 1C*), indicating that the peptidisc-embedded complexes remain stable in solution. Peak fractions were then imaged by negative-stain EM, which in all three cases showed a homogeneous and monodispersed particle population (*Figure 1—figure supplement 1*). Class averages calculated from these samples show the structural features expected for the RC core complex, the monomeric RC–LH1 complex, and the dimeric RC–LH1–PufX complex (*Figure 1D* and *Figure 1—figure supplement 1*). These results are consistent with the proposed tight fit of the peptidisc scaffold to the contour surface of the transmembrane domains of the stabilized proteins and complexes (*Carlson et al., 2018*).

## Peptidiscs preserve MsbA in its native conformation

To explore whether peptidisc-stabilized membrane proteins are suitable for high-resolution structure determination by single-particle cryo-EM, we used the ABC transporter MsbA as test specimen, as its structure was recently determined in different conformations in the context of a nanodisc (*Mi et al., 2017*).

We purified His-tagged MsbA by nickel-affinity purification and, because the protein is quite stable, we opted not to use the on-gradient reconstitution approach and instead reconstituted it into peptidiscs using the conventional on-beads method described before (*Figure 2—figure supplement 1A*; *Carlson et al., 2018*). The peak fractions from the nickel-affinity column were pooled and analyzed by SEC, which showed a sharp and symmetric peak for the peptidisc-embedded MsbA (*Figure 2—figure supplement 1B*) and by negative-stain EM, which showed monodispersed particles that were homogeneous in size (*Figure 2—figure supplement 2A*). Analysis of the ATPase activity of MsbA purified in dodecyl maltoside (DDM) and reconstituted into peptidiscs established that the protein was about six times more active in peptidisc than in detergent (*Figure 2—figure supplement 2B*), similar to what has been observed for MsbA reconstituted into lipid-based nanodiscs (*Mi et al., 2017*). Consistently, class averages of negatively stained MsbA in peptidiscs showed that all transporters assumed similar conformations, also seen in nanodiscs, in which the nucleotide-binding domains (NBDs) are in close proximity to each other (*Figure 2—figure supplement 2C*). None of the averages showed the detergent-induced conformation, seen in an X-ray crystal structure (*Ward et al., 2007*) and negative-stain EM averages of detergent-solubilized MsbA (*Mi et al., 2017*; *Moeller et al., 2015*), in which the NBDs are far apart. Even though double electron-electron resonance experiments have found that the NBDs of MsbA reconstituted into proteoliposomes can be distant from each other (*Zou et al., 2009*), because we only see separated NBDs for detergent-solubilized MsbA, which has a lower ATPase activity, but not for MsbA reconstituted into peptidiscs (*Figure 2—figure supplement 2C*) or nanodiscs (*Mi et al., 2017*), we believe that NBD separation is an artifact introduced by the detergent.

The structure of MsbA reconstituted into a peptidisc was then analyzed by single-particle cryo-EM. Samples were vitrified, imaged with a K2 Summit direct detector on a Titan Krios electron microscope and processed with RELION-3 (*Zivanov et al., 2018*; *Figure 2—figure supplements 3*, *5* and *6*). Image processing revealed subtle conformational variability in the distance between the two NBDs, as illustrated by the two final density maps at resolutions of 4.2 and 4.4 Å (*Figure 2A* and *Figure 2—figure supplement 6A*). The slight mobility of the NBDs in the nucleotide-free state is the likely reason for the somewhat limited resolution of our maps as well as the previously published cryo-EM map of MsbA in nanodiscs (also 4.2 Å) (*Mi et al., 2017*). The final model was

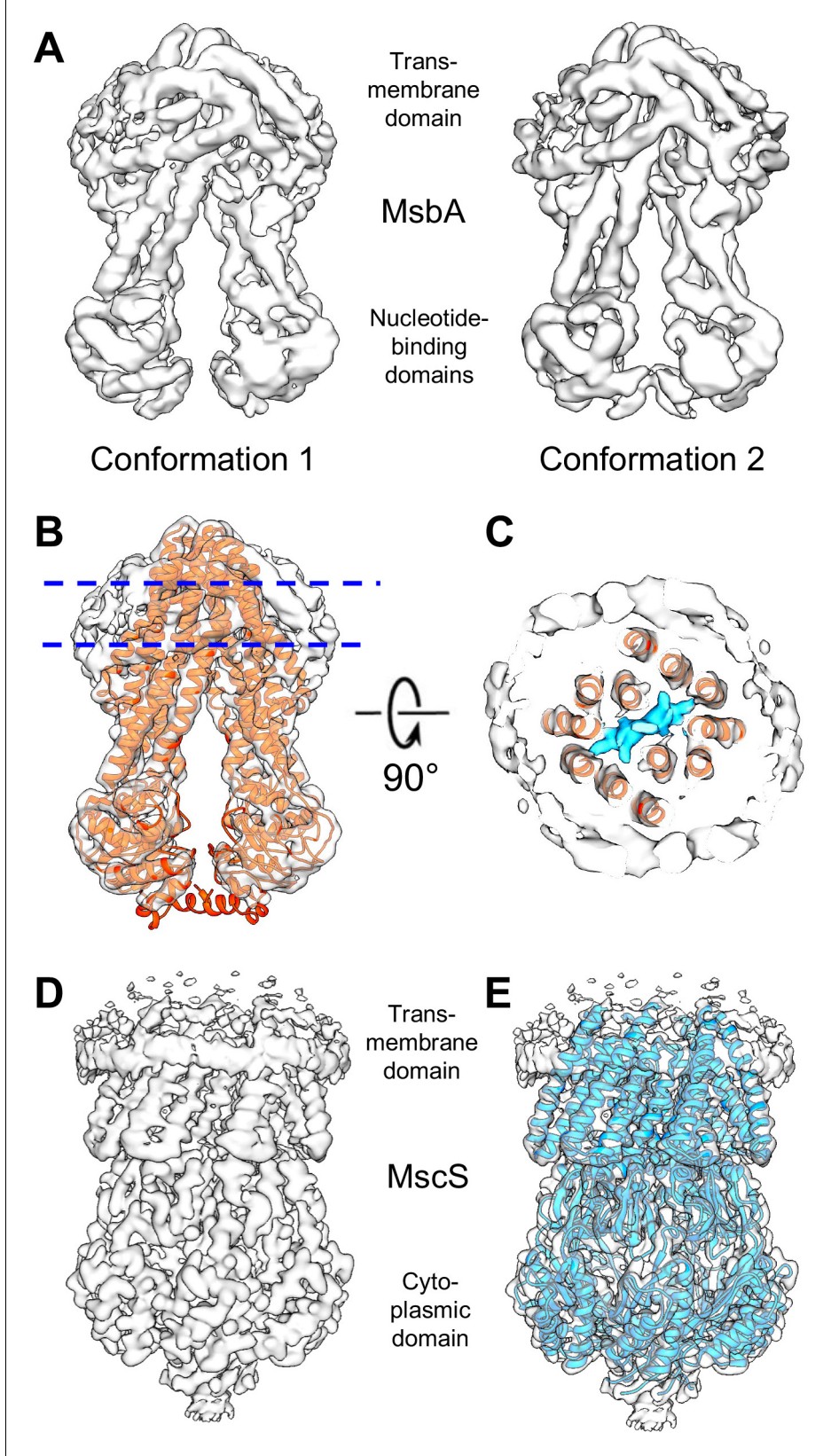

**Figure 2.** Cryo-EM maps for the *E. coli* lipid transporter MsbA and the *E. coli* mechanosensitive channel MscS reconstituted into peptidiscs. (**A**) Two density maps for MsbA in peptidiscs that differ slightly in the relative
*Figure 2 continued on next page*

*Figure 2 continued*

position of the two nucleotide-binding domains). The resolution of the density map of MsbA in conformation 1, in which the NBDs are slightly more separated, is 4.2 Å and that of the density map of MsbA in conformation 2 is 4.4 Å. (B) Density map of MsbA in conformation 1 with the cryo-EM structure of MsbA in nanodiscs refined into the map (PDB: 5TV4; *Adams et al., 2010*). (C) Section through the map of MsbA in conformation 1 at the position indicated in panel B. In addition to MsbA (red ribbon representation) and density for peptidisc peptides (light gray surface), the map reveals density for a bound lipopolysaccharide (LPS) molecule (cyan surface). (D) Density map of MscS in a peptidisc at a resolution of 3.3 Å. (E) Density map of MscS with the crystal structure refined into the map (PDB: 2OAU; *Bass et al., 2002*).

The online version of this article includes the following figure supplement(s) for figure 2:

**Figure supplement 1.** Purification of MsbA and MscS in peptidiscs.
**Figure supplement 2.** Characterization of MsbA and MscS in peptidiscs.
**Figure supplement 3.** Cryo-EM analysis of MsbA in peptidiscs.
**Figure supplement 4.** Cryo-EM analysis of MscS in peptidiscs.
**Figure supplement 5.** Local resolution of the cryo-EM maps of MsbA and MscS in peptidiscs.
**Figure supplement 6.** Characterization of the density maps of MsbA and MscS in peptidiscs.
**Figure supplement 7.** Comparison of MscS in different membrane mimetics.
**Figure supplement 8.** Comparison of density maps generated with and without imposing symmetry.

---

obtained by docking the cryo-EM structure of MsbA in nanodiscs into our 4.2 Å density map using Chimera (*Pettersen et al., 2004*), followed by refinement in PHENIX (*Adams et al., 2010*; *Figure 2B*). Like in the nanodisc-stabilized MsbA, our map showed clear density for LPS trapped in the transporter (*Figure 2C*), which was not visible in any of the X-ray crystal structures of detergent-solubilized MsbA . However, unlike the EM map of nanodisc-stabilized MsbA, in which the nanodisc is seen as a large amorphous density, the EM map of peptidisc-stabilized MsbA shows clear tubular densities surrounding the transmembrane domain of MsbA that represent the α-helical peptidisc peptides (see *Figure 2A* and below).

Together, these results show that peptidiscs are similarly effective in providing a membrane-like environment for membrane proteins as the MSP-based, lipid-containing nanodiscs and that membrane proteins reconstituted into peptidiscs are suitable for structure determination by single-particle cryo-EM.

## Peptidiscs allow cryo-EM structure determination of MscS to high resolution

Single-particle cryo-EM of MsbA in peptidiscs yielded density maps at a similar resolution as was achieved with MsbA in nanodiscs, but the resolution of these maps is somewhat limited, presumably due to the slight mobility of the NBDs in the nucleotide-free state. To establish whether peptidiscs also allow the structure of membrane proteins to be determined at higher resolution, we used MscS as test specimen, as its structure in nanodiscs has recently been determined to resolutions of 2.9 Å (*Rasmussen et al., 2019*) and 3.1 Å (*Reddy et al., 2019*).

His-tagged MscS was purified in DDM, reconstituted into peptidiscs using the on-beads method and further purified by SEC (*Figure 2—figure supplement 1C and D*). Negative-stain EM showed a monodispersed population of particles of homogeneous size, establishing that the sample was suitable for analysis by cryo-EM (*Figure 2—figure supplement 2D*). The sample was vitrified, imaged and processed in RELION-3 in the same way as MsbA (*Figure 2—figure supplement 4 to 6* ). The final map reached a resolution of 3.3 Å (*Figure 2D* and *Figure 2—figure supplement 6B*), close to the resolutions of 2.9 Å and 3.1 Å of the recently published cryo-EM maps of MscS reconstituted into nanodiscs (*Rasmussen et al., 2019*; *Reddy et al., 2019*). The resolution of our map is similar to that of the recently published single-particle cryo-EM reconstruction of MscS in DDM (3.4 Å) (*Reddy et al., 2019*), but our map shows better defined density for the loop connecting transmembrane helices 1 and 2 (*Figure 2—figure supplement 7A*). As MscS in nanodiscs, our map of MscS in peptidiscs shows the channel in the closed conformation, with perfect seven-fold symmetry that could be docked into our map using Chimera, followed by refinement in PHENIX (*Figure 2E*). Our final structure of MscS in peptidiscs is very similar to the published one of MscS in nanodiscs (*Figure 2—figure supplement 7B and C*). However, unlike the map of MscS in nanodiscs that resolves bound lipids, our map of MscS in

peptidiscs, like the X-ray crystallographic structure of detergent-solubilized MscS (*Bass et al., 2002*), does not show any density representing lipids associated with MscS.

## The arrangement of the peptidisc peptides around MsbA and MscS is different

Cryo-EM maps of membrane proteins reconstituted into nanodiscs typically show the nanodisc as an amorphous band of density. This is due to the fact that the lipids encircled by the MSPs, other than ones that are tightly associated with the target protein, are mobile, and that the MSPs do not assume unique positions with respect to the incorporated protein. As a result, all structural detail of the nanodisc are being averaged out, which is also true for detergent micelles surrounding a membrane protein. In contrast, the recent cryo-EM map of a fungal mitochondrial calcium transporter stabilized with saposin A showed clear density for the individual saposin A molecules, revealing how they are arranged around the protein (*Nguyen et al., 2018*). We therefore analyzed the non-protein density in our cryo-EM maps of peptidisc-stabilized MsbA and MscS to see whether it is possible to deduce the arrangement of the peptidisc peptides. To assess whether imposing symmetry during image processing introduced artificial features in the density representing the peptidisc peptides, we also processed the datasets without imposing symmetry. As expected, the resulting maps are noisier and have lower resolution, but the density representing the peptidisc peptides is very similar to that seen in the symmetrized maps (*Figure 2—figure supplement 8*).

In the case of MsbA, we identified a total of 15 and 21 helical segments in the maps of conformation 1 and 2, respectively (*Figure 3A and B*). The size of the identified segments is found to vary greatly in length. In both maps, only a single segment is seen with a length approaching the expected length if the entire peptidisc peptide adopts an α-helical conformation (~55 Å). Most other helical segments vary in length, ranging from 16 to 42 Å, and a few segments are very short (6.5 to 11 Å; *Figure 3*). The differing number of segments identified in the two maps somewhat exaggerates the actual difference in the MsbA surface covered by peptides in the two maps, because longer segments in one map occasionally overlap with more than one shorter segment in the other map. For example, the longest helical segment in the map of conformation 1 overlaps with three shorter segments in the map of conformation 2 (*Figure 4A*). Overall, most of the helical segments overlap completely or at least partially between the two maps, as can be seen on the front face of the MsbA dimer (*Figure 4B*, left panel). However, the side face of the MsbA dimer reveals differences between the surface coverage of the transmembrane domain by the peptidisc peptides, with the map of conformation 2 showing two helical segments that are not seen in the map of conformation 1 (*Figure 4B*, right panel; red arrows). In addition to the length, the tilt of the helical segments on the MsbA surface also varies. Most segments run almost parallel to the membrane plane, with tilt angles of less than ~15° for 8 and 14 segments for the maps of conformations 1 and 2, respectively. However, there are also four segments in both maps with higher tilt angles that range from ~40° to ~50°.

In the case of MscS, the peptide densities are organized very differently from those seen in the MsbA maps (*Figure 3C*). Of the 16 identified helical segments, two are long (~36 Å), three are short (~6, 13 and 16 Å), and all others have a length ranging from 19 to 32 Å. Importantly, all helical segments run essentially parallel to the membrane plane and form a continuous band of helical segments, surrounding the transmembrane domain of MscS at a level corresponding to the cytoplasmic leaflet of the membrane. A second continuous band is also present at the level corresponding to the periplasmic leaflet of the membrane, but these helical segments appear shorter and less well defined in our map. Thus, unlike in the case of MsbA, the arrangement of the peptidisc peptides surrounding the MscS channel is reminiscent of the double-belt model proposed for how MSPs encircle the lipids in nanodisc (*Bibow et al., 2017*).

## Discussion

Stabilizing membrane proteins in solution for structural and functional studies remains an important area of research, which has gained further importance with the recent advances in single-particle cryo-EM that resulted in a tremendous increase in the pace at which membrane protein structures can now be determined. Preparing well-behaved samples has thus become one of the time-limiting steps for structure determination of membrane proteins. Because peptidiscs, unlike detergents, do not affect the structure nor decrease the stability of membrane proteins and because reconstitution

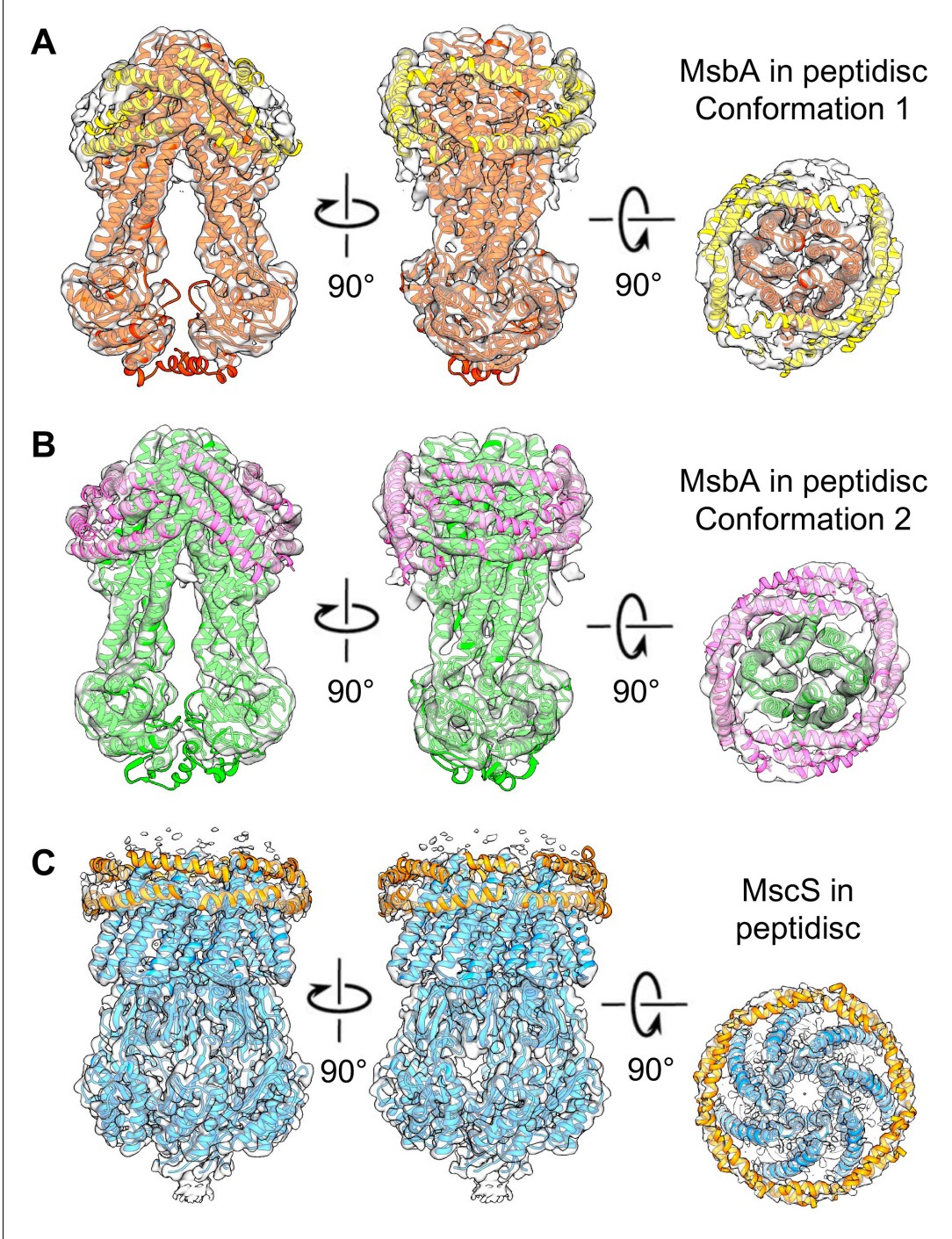

**Figure 3.** The peptidisc densities surrounding MsbA and MscS. Different views of the density maps of (**A**) MsbA in conformation 1 (red ribbon representation), (**B**) MsbA in conformation 2 (green ribbon representation), and (**C**) MscS (blue ribbon representation). Densities consistent with helical segments surrounding the membrane proteins and representing peptidisc peptides were fit with poly-alanine models (yellow ribbon representation).

of membrane proteins into peptidiscs requires little optimization, peptidiscs have emerged as a promising new tool to stabilize membrane proteins in solution. Here, we present the on-gradient method that combines gentle reconstitution with a purification step and may be particularly well suited to reconstitute labile membrane-protein complexes into peptidiscs. In addition, our high-resolution cryo-EM structures of MsbA and MscS reveal very different arrangements of the peptidisc

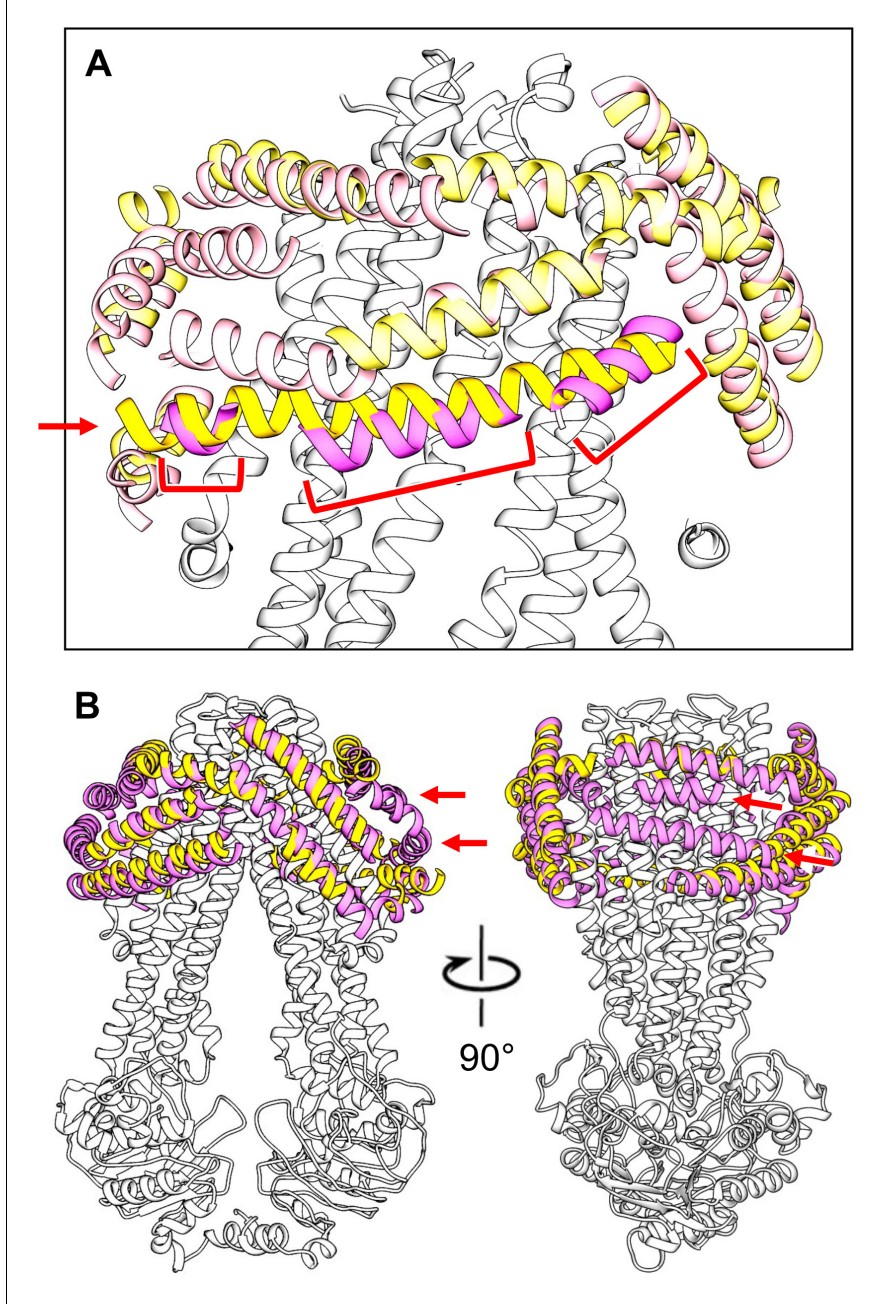

**Figure 4.** Comparison of modeled peptide helices surrounding MsbA in conformations 1 and 2. (**A**) The longest helical segment found in the peptidisc density surrounding MsbA in conformation 1 (yellow helix indicated with red arrow) overlaps with three helical segments found in the peptidisc density surrounding MsbA in conformation 2 (magenta helices indicated with red brackets). (**B**) Orthogonal views of the superimposition of the modeled peptidisc peptide helices in the map of MsbA in conformation 1 (yellow helices) with those in the map of MsbA in conformation 2 (magenta helices) showing that many helices in the two maps overlap. The map of MsbA in conformation 2 shows two helical segments that are not present in the map of MsbA in conformation 1 (indicated by red arrows).

peptides, thus revealing the structural basis for why peptidiscs are an excellent and probably universal tool to stabilize membrane proteins outside of biological membranes.

In our original report, we showed that detergent-solubilized membrane proteins can be reconstituted into peptidiscs by the on-bead method, in which the protein is immobilized on beads and the

detergent buffer is replaced with buffer containing the peptidisc peptide before the protein is eluted from the beads (*Carlson et al., 2018*), a method now named PeptiQuick (*Saville et al., 2019*). The novel on-gradient method we present here combines membrane protein reconstitution into peptidiscs with a purification step, because the reconstituted target protein will form a defined band in the gradient, whereas other proteins will localize to different positions in the gradient. In particular, small degradation products, excess peptides and detergent micelles stay at the top of the gradient, whereas aggregates migrate further towards the bottom of the centrifuge tube. The location of the reconstituted target protein in the gradient can always be identified by gradient fractionation and analyzing the fractions by SDS-PAGE. However, as seen with the RC complexes, colored proteins are easily seen as discrete bands in the gradient (*Figure 1B*). Therefore, target proteins could be expressed with a fluorescent tag, such as green fluorescent protein or mCherry, to simplify their harvest from the gradient. Alternatively, the peptidisc peptide could be labeled with a fluorophore and mixed in with unlabeled peptide used for reconstitution, which would be a universally applicable approach that does not require modification of the target protein. Although more work will be needed to fully establish this approach, initial attempts look very promising (*Figure 1— figure supplement 2*).

We believe that the on-gradient method is a particularly gentle and simple way to reconstitute membrane proteins into peptidiscs, because it does not require immobilization and elution of the target protein from a resin, and because sugars, such as glycerol and sucrose, are known to have a stabilizing effect on proteins and macromolecular complexes (*Ruan et al., 2003*). As a natural extension of the method, on-gradient peptidisc reconstitution could be combined with the GraFix approach, in which the sugar gradient also contains a gradient of glutaraldehyde to chemically fix fragile membrane-protein complexes (*Kastner et al., 2008*). Considering that eight of the 37 residues of the peptidisc peptide are lysines, this extension should also introduce crosslinks between the individual peptides, which would rigidify the scaffold and further stabilize the target complex (which may form additional crosslinks with the scaffold). This approach would thus have the potential to stabilize even the most labile membrane-protein complexes.

The resolutions of the cryo-EM structures we obtained for peptidisc-stabilized MsbA and MscS are similar to those reported earlier for the same proteins embedded in nanodiscs. Thus, the ease that peptidiscs provide for reconstituting membrane proteins does not compromise the resolution that can be achieved. In addition, the structures of MsbA and MscS in peptidiscs are indistinguishable from those of the two proteins in nanodiscs, demonstrating that the peptidisc scaffold provides a similar stabilizing effect as the MSP-based, lipid-containing nanodiscs. In the case of MsbA, the protein in peptidiscs has a higher ATPase activity than in detergent, as was seen for the protein in nanodiscs, and it does not adopt the artifactual conformation with widely separated NBDs observed with detergent-solubilized protein (*Mi et al., 2017*; *Moeller et al., 2015*; *Ward et al., 2007*). Similarly, MscS in peptidiscs adopts the expected closed channel conformation, showing that the peptidisc environment does not exert major strain on this mechanosensitive channel. Interestingly, the peptidisc environment also preserves the LPS cargo bound to MsbA as was observed for this transporter reconstituted into nanodiscs (*Mi et al., 2017*). In the case of MscS, however, the map does not reveal density that could be assigned to bound lipids as seen in the structure of MscS reconstituted into nanodiscs (*Rasmussen et al., 2019*; *Reddy et al., 2019*). These findings suggest that peptidiscs cannot preserve lipids that are only loosely associated with the protein or whose presence depends on the physicochemical properties of a true lipid bilayer, which is provided by MSP-based lipid nanodiscs. However, peptidiscs do preserve lipids that are tightly bound to membrane proteins, which are also the most likely ones to affect membrane protein function.

Analysis of the density representing the peptidisc in our cryo-EM maps provides the first glimpse of how the peptides cover the hydrophobic domain of membrane proteins. Their arrangement appears to be in between that of detergent micelles and nanodiscs on the one hand, and that of saposin A on the other. Instead of an amorphous density surrounding the target protein typically seen for detergent micelles and nanodiscs, our maps of peptidisc-embedded MsbA and MscS show well-defined densities consistent with α-helical peptides. However, these densities have different lengths and represent only parts of the entire peptidisc peptides, which is different from a cryo-EM map of a fungal mitochondrial calcium transporter obtained with saposin A. In the latter, six saposin A molecules surround the transporter, which could all be modeled almost completely as poly-alanine α-helices (*Nguyen et al., 2018*). However, to date this is the only structure of a membrane protein

that has been determined at near-atomic resolution with this membrane mimetic. A potential reason may be that saposin A is only compatible with membrane proteins whose hydrophobic belt can accommodate a multiple of the helix-loop-helix structure of saposin A. In contrast, the peptidisc peptide consists of two short amphipathic helical segments separated by a proline residue. This shorter peptide design probably allows the scaffold to adapt to a variety of transmembrane domains, including those of β-barrel proteins (*Saville et al., 2019*). This notion is supported by saposin A being arranged in a well-organized tilted picket fence arrangement in the high-resolution structure of the mitochondrial calcium transporter (*Nguyen et al., 2018*), whereas the organization of the peptidisc peptide surrounding MsbA and MscS varies greatly. In the case of MsbA, the helical peptide segments are tilted at different angles with respect to the membrane plane, whereas the peptides surrounding MscS run essentially parallel to the membrane. The variable way in which peptidisc peptides can adapt to the hydrophobic domain of membrane proteins is also reflected in the overall length of the densities representing the peptides, since most are shorter than the expected length of the peptide. Since single-particle cryo-EM maps result from the averaging of thousands of individual particle images, density for the peptidisc peptides only represents the most commonly assumed positions, and even these densities are weaker than those corresponding to the transmembrane helices of the incorporated target protein (*Figure 2—figure supplement 5*). The weaker density and the many densities that are shorter than the peptidisc peptide thus suggest that the peptides can cover the hydrophobic belt of the target membrane protein in a variety of ways. This conclusion is supported by our earlier native mass spectrometry and peptide quantitation analysis, which suggested that proteins are not necessarily stabilized by a defined number of peptidisc peptides (*Carlson et al., 2018*). We therefore conclude that the different ways in which the peptidisc peptide can assemble around transmembrane domains renders this membrane mimetic ideally suited to stabilize many if not all membrane proteins in the absence of a lipid bilayer.

The fact that at least some regions of the peptidisc peptides are well ordered provides an additional motivation for using peptidiscs as membrane mimetic. Detergent micelles and nanodiscs are structurally very heterogeneous. Therefore, even though they increase the size of the target membrane protein and thus make the particles easier to see and to pick, they also bury the transmembrane domain in an amorphous density, thus preventing it from contributing signal that could be exploited for alignment, 2D and 3D classification of the particles, especially at early stages of the processing when the resolution of the map is still low. In contrast, while peptidiscs also increase the size of the target membrane protein, the regions of the peptidisc peptides that assume defined positions on the protein also contribute signal that will help in the alignment, 2D and 3D classifications of the particles, similar, for example, to Fab fragments bound to target proteins (*Wu et al., 2012*). The signal of the ordered peptidisc peptide regions should thus aid in image processing and is therefore an additional advantage of using peptidiscs over detergents or nanodiscs.

## Materials and methods

**Key resources table**

| Reagent type (species) or resource | Designation | Source or reference | Identifiers | Additional information |
|---|---|---|---|---|
| Gene (*Escherichia coli*) | mscS | *Bass et al., 2002* | 83333 (NCBI) | Kind gift from Dr. Douglas Rees |
| Strain, strain background (*Escherichia coli*) | BL21(DE3) | Agilent (Stratagene) | #200131 | Competent cells |
| Strain, strain background (*Escherichia coli*) | C43 (DE3) | *Miroux and Walker, 1996* | | Kind gift from Dr. Bruno Miroux |
| Strain, strain background (*Rhodobacter sphaeroides*) | RCx^R | *Jun et al., 2018* | ΔpuhA, ΔpufQBALMX, Δrshl, ΔppsR | Kind gift from Dr. Tom Beatty |

*Continued on next page*

*Continued*

| Reagent type (species) or resource | Designation | Source or reference | Identifiers | Additional information |
|---|---|---|---|---|
| Recombinant DNA reagent | pET28b (Plasmid) | Addgene | 69865–3 | - |
| Recombinant DNA reagent | pIND4-RC (Plasmid) | *Carlson et al., 2018*; *Jun et al., 2018* | | |
| Recombinant DNA reagent | pIND-RC1 (Plasmid) | *Jun et al., 2018*; *Carlson et al., 2018* | | |
| Recombinant DNA reagent | pBAD-His6-MsbA (Plasmid) | *Carlson et al., 2018* | | |
| Peptide, recombinant protein | Peptidiscs | Peptidisc Biotech. | Bulk-Peptidisc Glo-Peptidisc | - |
| Chemical compound, drug | Uranyl formate | Pfaltz and Bauer | Cat# U01000 | - |
| Chemical compound, drug | LB broth miller powder | Affymetrix/USB | 4340023 | Media |
| Chemical compound, drug | kanamycin | Teknova | K2150 | antibiotic |
| Chemical compound, drug | Isopropyl β-D-thiogalactoside (IPTG) | Gold Biotechnology | Cat# I2481C | - |
| Chemical compound, drug | Roche cOmplete, EDTA-free protease inhibitor | Sigma-Aldrich | Cat# 11873580001 | - |
| Chemical compound, drug | 1,4-dithiotreitol (DTT) | Gold Biotechnology | Cat# DTT25 | - |
| Chemical compound, drug | Tris base | Chemcruz | sc-3715B | - |
| Chemical compound, drug | N,N-dimethyldodecylamine N-oxide | Sigma | Cas# 1643-20-5 | |
| Chemical compound, drug | Triton X-100 | BioShop | Cat# TRX 777.500 | |
| Chemical compound, drug | n-dodecyl-β-D-maltoside | Anatrace | Cas# 69227-93-6 | |
| Chemical compound, drug | octyl-β-D-glucoside | Anatrace | Cas# 29836-26-8 | |
| Chemical compound, drug | Arabinose | GoldBio | Cat# A-300–1 | |
| Chemical compound, drug | Imidazole | Bioshop | Cat# IMD508.1 | |
| Chemical compound, drug | Acrylamide | BioShop | Cat# ACR005.502 | |
| Chemical compound, drug | Ampicillin | Bioshop | Cat# AMP201.25 | |
| Chemical compound, drug | Malachite Green | Sigma | Cas# 569-64-2 | |
| Chemical compound, drug | Sucrose | Bioshop | Cat# SUC507.5 | |
| Software, algorithm | RELION-3.0 | *Zivanov et al., 2018* | http://www2.mrc-lmb.cam.ac.uk/relion | |
| Software, algorithm | MotionCor2 | *Zheng et al., 2017* | http://msg.ucsf.edu/em/software/motioncor2.html | - |
| Software, algorithm | CtfFind4.1.8 | *Rohou and Grigorieff, 2015* | http://grigoriefflab.janelia.org/ctffind4 | - |

*Continued on next page*

*Continued*

| Reagent type (species) or resource | Designation | Source or reference | Identifiers | Additional information |
|---|---|---|---|---|
| Software, algorithm | Gautomatch | N/A | https://www.mrc-lmb.cam.ac.uk/kzhang/Gautomatch | - |
| Software, algorithm | SPIDER2 | *Yang et al., 2017* | http://sparks-lab.org/yueyang/server/SPIDER2/ | - |
| Software, algorithm | COOT | *Emsley et al., 2010* | http://www2.mrc-lmb.cam.ac.uk/personal/ pemsley/coot | - |
| Software, algorithm | PHENIX | *Adams et al., 2010* | https://www.phenix-online.org | - |
| Software, algorithm | Chimera | *Pettersen et al., 2004* | https://www.cgl.ucsf.edu/chimera/download.html | - |
| Software, algorithm | Serial EM | *Mastronarde, 2005* | http://bio3d.colorado.edu/SerialEM | - |
| Other | R 2/2 400 mesh Cu Holey carbon grids | Quantifoil | Q450CR2 | Cryo-EM grids |
| Other | R 1.2/1.3 400 mesh Cu Holey carbon grids | Quantifoil | Q4100CR1.3 | Cryo-EM grids |

## Reagents

Peptidisc peptides were obtained from Peptidisc Biotech. Tryptone, yeast extract, Tris-base, imidazole, NaCl, dithiothreitol (DTT), acrylamide 40%, bis-acrylamide 2% and TEMED were obtained from Bioshop, Canada. Isopropyl β-D-1-thiogalactopyranoside (IPTG), ampicillin, kanamycin and arabinose were purchased from GoldBio, EDTA-free protease inhibitor cocktail from Roche, n-dodecyl-β-D-maltoside (DDM) and octyl-β-D-glucoside (β-OG) from Anatrace, and N,N-dimethyldodecylamine N-oxide (LDAO) and Triton X-100 from Sigma. Superdex 200 HR and Superose 6 resins were obtained from GE Healthcare. Most other chemicals were purchased from Fisher Scientific.

## Expression and purification of RC complexes

The RC core and RC-LH1 complexes were purified as described before (*Jun et al., 2018*). Briefly, RC core complex with a C-terminally His-tagged H subunit was expressed in *Rhodobacter sphaeroides* strain RCx$^R$ (Δ*puhA*, Δ*pufQBALMX*, Δ*rshI*, Δ*ppsR*) using plasmid pIND4-RC, and RC–LH1 complex was expressed in *R. sphaeroides* strain RCx$^R$ using plasmid pIND-RC1. A 10 mL pre-culture in RLB medium (Luria Broth with 810 µM MgCl$_2$ and 510 µM CaCl$_2$) supplemented with 25 µg/mL kanamycin was transferred into 100 mL of the same kanamycin-supplemented RLB medium, grown overnight at 30℃, and then transferred into 1 L of the same medium. After 8 hr at 30℃, protein production was induced with 1 mM IPTG and the cells were grown for an additional 16 hr. During growth and purification, light exposure was kept to a minimum. Cells were harvested by low-speed centrifugation (10,000 x*g* at 4℃ for 6 min), resuspended in Buffer A (50 mM Tris-HCl, pH 7.9, 200 mM NaCl) and lysed with a microfluidizer (15,000 psi). Unbroken cells and cell debris were removed by low-speed centrifugation (10,000 x*g* at 4℃ for 6 min), and the supernatant was incubated with 1% (w/v; final concentration) LDAO overnight at 4℃ with shaking. After removal of insoluble material by ultracentrifugation (200,000 x*g* at 4℃ for 30 min), the supernatant was supplemented with 10 mM imidazole and applied to a nickel-affinity column. The column was washed overnight at 4℃ with 5 mM imidazole in Buffer A containing 0.03% LDAO (Buffer A $_{LDAO}$), and bound protein was eluted with 600 mM imidazole in the same buffer. The peak fractions were concentrated using a centrifugal filter (Amicon 30 kDa cut-off for RC core and 100 kDa cut-off for RC–LH1), then subjected to size-exclusion chromatography (SEC) using a Superdex 200 10/300 GL column equilibrated with Buffer A $_{LDAO}$. Fractions were collected in 500 µL aliquots, and stored in the dark at 4℃ until use.

The *R. sphaeroides* cells enriched for the RC–LH1–PufX complex were generously provided by Dr. Tom Beatty at the University of British Columbia (*Abresch et al., 2005*). The cells were harvested by low-speed centrifugation, resuspended in Buffer A and lysed with a microfluidizer (15,000 psi). Unbroken cells and cell debris were removed by low-speed centrifugation. Cell membranes were isolated by centrifugation at 150,000 x*g* for 45 min at 4℃, resuspended in Buffer A to a concentration

of 5 mg/mL, and solubilized with 0.5% (w/v) sodium cholate and 4% (w/v) OG for 30 min at room temperature with stirring. After removal of insoluble material (150,000 x$g$ for 45 min at 4°C), the supernatant was applied to a nickel-affinity column. The bound complex was washed with 40 column volumes (CV) of Buffer A containing 0.2% sodium cholate and 0.02% DDM. The complex was eluted with 300 mM imidazole in Buffer A $_{DDM}$. The peak fractions were pooled and concentrated to 5–10 mg/mL on a 100 kDa cut-off centrifugal filter (Amicon). The concentrated protein was layered onto a linear 15–35% sucrose gradient and centrifuged at 120,000 x$g$ for 48 hr at 4°C. The gradient was manually harvested from the top into 1 mL fractions. Fractions containing the RC–LH1–PufX complex were pooled, concentrated and subjected to SEC on a Superose 6 10/300 column equilibrated with Buffer A containing 0.02% DDM (Buffer A $_{DDM}$). The fractions containing the complex were pooled and stored in the dark at 4°C until use.

## On-gradient reconstitution of RC complexes into peptidiscs

For on-gradient reconstitution, 1 mL of a 1:1.6 (w/w) mixture of target membrane protein (at a concentration ranging from 0.5 to 5 mg/mL) and peptidisc peptide were overlaid onto a 12 mL linear sucrose gradient (5–20% in Buffer A). After centrifugation at 210,000 x$g$ for 15 hr at 4°C in an SW41 rotor (Beckman Coulter), the gradient was manually fractionated from the top into 1 mL aliquots. The fractions were analyzed by SDS-PAGE. Peak fractions were collected and concentrated on a 30 kDa cut-off centrifugal filter (Amicon). The concentrate was injected onto a Superose 6 10/300 GL column pre-equilibrated in Buffer A at a flowrate of 0.5 mL/min. Fractions were collected in 500 µL aliquots and analyzed by SDS-PAGE.

## Expression, purification and on-bead reconstitution of MsbA and MscS into peptidiscs

The N-terminally His-tagged MsbA in pET-28 vector was transformed into *E. coli* strain C43. Cells were grown in autoinduction medium supplemented with 200 µg/mL kanamycin for 3 hr at 37°C, followed by overnight incubation at room temperature. Cells were harvested by centrifugation (10,000 x$g$, 6 min, 4°C), resuspended in Buffer A, and lysed using a high-pressure microfluidizer (Microfluidics) at 15,000 psi. Unbroken cells and cell debris were removed by low-speed centrifugation (10,000 x$g$, 6 min, 4°C), and membranes were then pelleted by high-speed centrifugation (100,000 x$g$, 45 min, 4°C). The membranes were resuspended in Buffer A and solubilized with 1% (w/v) DDM at a final protein concentration of 5 mg/mL for 1 hr at 4°C with stirring. After removal of insoluble material by high-speed centrifugation (200,000 x$g$, 30 min, 4°C), imidazole was added to the supernatant to a final concentration of 5 mM and applied to 5 mL of Ni$^{2+}$-NTA chelating resin. The resin was washed with 20 CV of 5 mM imidazole in Buffer A $_{DDM}$. MsbA was eluted with 2 CV of 600 mM imidazole in Buffer A $_{DDM}$ and the peak fractions were pooled, concentrated to a concentration of 5 mg/mL on a 100 kDa cut-off Amicon filter, and subjected to SEC on a Superdex 200 10/300 GL column equilibrated in Buffer A $_{DDM}$. Purified MsbA was concentrated to 10–15 mg/mL and stored at −80°C.

On-bead reconstitution into peptidiscs was carried out as described (*Carlson et al., 2018*). Briefly, 50 mL of crude membrane from MsbA-expressing *E. coli* cells at 5 mg/mL were solubilized with 1% DDM in Buffer A. Insoluble material was removed by ultracentrifugation (100,000 x$g$, 45 min, 4°C), and the supernatant was applied to 5 mL of Ni-NTA chelating resin (Goldbio). The resin was washed with 20 CV of 5 mM imidazole in Buffer A $_{DDM}$, and the beads were collected by centrifugation (2500 x$g$, 10 min, 4°C). After removing excess buffer, the beads were resuspended in 10 CV of Buffer A containing peptidisc peptide (1 mg/mL). Excess peptide was washed away with 5 CV of Buffer A. The protein was eluted with 600 mM imidazole in Buffer A. Peak fractions were pooled, concentrated to 10 mg/mL with a 100 kDa cut-off Amicon filter, and injected onto a Superdex 200 GL 10/300 column in Buffer A at a flow rate of 0.25 mL/min. Fractions were collected and analyzed by SDS-PAGE.

Full-length *E. coli* MscS was cloned into pET-28b(+) vector with an N-terminal His tag and used to transform *E. coli* strain BL21 (DE3). Cells were grown at 37°C in LB medium containing 50 µg/mL kanamycin. When the culture reached an OD600 of 0.6, cells were induced with 1 mM IPTG. After another 4 hr at 37°C, cells were harvested by centrifugation (4000 x$g$, 30 min, 4°C), quick-frozen in liquid nitrogen, and stored at −80°C until use. Cell pellets were resuspended for 1 hr at 4°C in lysis

buffer (40 mM Tris-HCl, pH 7.9, 500 mM NaCl, 1% Triton X-100 (w/v)) complemented with 1 tablet of EDTA-free protease inhibitor cocktail. Cells were lysed by sonication with a probe sonicator using an amplitude value of 40% for 15 min (cycles of 3 s ON and 8 s OFF). After centrifugation at 300,000 rpm using a 70 Ti rotor (Beckman Coulter) for 20 min at 4℃, the supernatant was loaded onto a Ni-NTA agarose column (QIAGEN) equilibrated with 10 CV of 1% Triton X-100 and 40 mM imidazole in 40 mM Tris-HCl, pH 8.0, 500 mM NaCl, and incubated for 1 hr at 4℃. After washing with 20 CV of 0.02% DDM in TS Buffer (50 mM Tris-HCl, pH 8.0, 50 mM NaCl), the column was washed with 20 CV of Buffer A containing peptidisc peptide (1 mg/mL in). After another 10 min incubation at 4℃, the column was further washed with 40 mM imidazole in TS Buffer. The protein, now reconstituted into peptidiscs, was eluted with 250 mM imidazole in TS Buffer. Peak fractions were combined and subjected to SEC on a Superdex 200 10/300 GL column equilibrated in Buffer B (40 mM Tris-HCl, pH 7.9, 150 mM NaCl). Peak fractions were pooled, concentrated to 1 mg/mL using centrifugal filters (100 kDa cut-off; Millipore), and stored at −80℃. The purity of the protein was assessed by SDS-PAGE and negative-stain EM.

## MsbA ATPase activity assay

Purified MsbA either in 0.02% DDM or reconstituted into peptidisc was assayed for ATPase activity using the malachite green protocol previously described (*Lanzetta et al., 1979*). Briefly, 1 μM of MsbA was incubated at 37℃ in translocation buffer (50 mM Tris-HCl, pH 7.8, 50 mM NaCl, 50 mM KCl, 10 mM $MgCl_2$, 1 mM DTT, and 0.02% DDM when needed) containing 1 mM ATP. Sample aliquots (5 μL) were mixed with 500 μL of activated Malachite Green and 0.5% Triton X-100 over a time course of 10 min. Light absorption was measured at 660 nm and activity was calculated using first-order rate kinetics.

## Negative-stain electron microscopy

RC complexes and MsbA reconstituted into peptidiscs were negatively stained with 0.7% uranyl formate as described (*Ohi et al., 2004*). Samples were imaged with a Philips CM10 electron microscope equipped with a tungsten filament and operated at 100 kV. All images were recorded on an AMT XR16L-ActiveVu charge-coupled device (CCD) camera using a nominal magnification of 52,000x, corresponding to a calibrated pixel size of 2.7 Å/pixel, and a defocus of −1.5 μm. For RC core complex, 42 images were recorded, 70 images for the RC–LH1 complex, 95 images for the RC–LH1–PufX complex, and 27 images for MsbA. Particles were manually picked with EMAN2 and windowed into individual images, yielding 13,177 90 × 90 pixel images for the RC core complex, 10,861 120 × 120 pixel images for the RC–LH1 complex, 10,653 180 × 180 pixel images for the RC–LH1–PufX complex, and 14,976 110 × 110 pixel images for MsbA. The particle images were centered, normalized and classified into 100 groups using *K*-means classification procedures implemented in SPIDER (*Frank et al., 1996*).

## Cryo-electron microscopy

For peptidisc-embedded MsbA, 3.5 μL aliquots at 1.3 mg/mL were applied to glow-discharged Quantifoil R2/2 400 mesh Cu holey carbon grids. Grids were blotted for 3.5 s at 4℃ with ~90% humidity and, after a waiting time of 10 s, plunge-frozen in liquid ethane using a Vitrobot Mark IV (Thermo Fisher Scientific). For peptidisc-embedded MscS, 3.5 μL aliquots at 0.07–0.12 mg/mL were applied to glow-discharged Quantifoil R1.2/1.3 400 mesh Cu holey carbon grids covered with a homemade thin carbon film. Grids were blotted for 0.5 s at 4℃ with 80–90% humidity and then plunge-frozen in liquid ethane using a Vitrobot Mark IV.

Cryo-EM data were collected using SerialEM on a 300-kV Titan Krios electron microscope (Thermo Fisher Scientific) in the Cryo-EM Resource Center at The Rockefeller University, equipped with a K2 Summit direct electron detector (Gatan) in super-resolution counting mode at a nominal magnification of 29,000x, corresponding to a calibrated super-resolution pixel size of 0.5 Å/pixel. Exposures of 10 s were dose-fractionated into 40 frames (250 ms per frame), with a dose rate of 8 electrons/pixel/s (~2 electrons/Å$^2$/frame), resulting in a total dose of 80 electrons/Å$^2$. The defocus range was varied from −1.0 to −2.0 μm for MsbA, and from −1.5 to −3.5 μm for MscS.

## Image processing

All movie frames were corrected with a gain reference collected during the EM session. UCSF MotionCor2 (*Zheng et al., 2017*) was used with 2x binning for motion correction and dose weighting. The contrast transfer function (CTF) was estimated using CTFFIND 4.1.8 (*Rohou and Grigorieff, 2015*). Particles were automatically picked with Gautomatch (https://www.mrc-lmb.cam.ac.uk/kzhang/Gautomatch/).

For peptidisc-embedded MsbA, 527,904 particles were picked from 1172 images and extracted into 256 × 256 pixel images. After 4x binning, the images were subjected to 2D classification into 100 classes in RELION-3 (*Zivanov et al., 2018*). Classes producing poor averages were discarded. The remaining 363,126 particle images were used at the original size for further classification and refinement in RELION-3. An initial 3D map, generated with eight classes showing different views of the protein, was C2-symmetrized and used as reference for 3D classification into six classes. Two classes showed the best structural detail but represented slightly different conformations, with the NBDs either close together (conformation 1) or slightly separated (conformation 2). Independent refinement and post-processing of these two classes yielded maps at resolutions of 5.5 Å and 5.8 Å, respectively. After CTF refinement and Bayesian polishing, the maps reached resolutions of 4.2 Å and 4.4 Å. All 3D classification and refinement steps were carried out with C2 symmetry imposed. To assess the effect of symmetrization on the density representing the peptidisc peptides, the 363,126 selected particle images were subjected to 3D classification into six classes without imposing symmetry, yielding two classes similar to those representing conformations 1 and 2 when C2 symmetry was imposed. These two classes, containing 124,691 and 84,079 particles, respectively, were independently refined without symmetry, resulting in maps at resolutions of 4.5 and 4.7 Å after post-processing.

For peptidisc-embedded MscS, first 218,672 particles were picked without templates from 1056 micrographs, extracted into 180 × 180 pixel images and subjected to 2D classification into 100 classes in RELION-3. Three averages showing the protein in different orientations were selected as templates to repick the images. The resulting 262,563 particles were extracted into 240 × 240 pixel images and subjected to another 2D classification into 100 classes. After discarding the classes that showed poor averages, the remaining 128,940 particles were used to generate an initial map imposing C7 symmetry, which was also imposed in all subsequent processing steps. A 3D classification into six classes produced four maps that showed more detailed structural features than the other two classes. The corresponding 99,883 particles were combined, refined and post-processed. After Bayesan polishing and additional refinement, the final map reached a resolution of 3.3 Å. In parallel, the 128,940 selected particles were also subjected to 3D classification into six classes without imposing symmetry. The class with the clearest features, containing 65,653 particles, was selected and subjected to 3D refinement. After polishing, the map was further refined and post-processed, all without imposing symmetry, yielding a final map at a resolution of 3.9 Å.

We used the command 'phenix.find_helices_strands' in PHENIX (*Adams et al., 2010*) to identify putative positions for α-helical segments and to build poly-alanine helices for all segments that were not accounted for by the protein models.

## Acknowledgements

We thank Professor Tom Beatty (University of British Columbia) for providing *R. sphaeroides* cells enriched in the RC–LH1–PufX complex and Mark Ebrahim and Johanna Sotiris for help with grid screening and data collection at the Evelyn Gruss Lipper Cryo-Electron Microscopy Resource Center at The Rockefeller University. Work in the FDVH laboratory was supported by a Discovery Grant for the Natural Sciences and Engineering Research Council of Canada.

## Additional information

### Competing interests

Franck Duong Van Hoa: is the scientific founder of Peptidisc Biotech. The other authors declare that no competing interests exist.

## Funding

| Funder | Grant reference number | Author |
|---|---|---|
| Natural Sciences and Engineering Research Council of Canada | RGPIN-2016-04241 | Franck Duong Van Hoa |

The funders had no role in study design, data collection and interpretation, or the decision to submit the work for publication.

## Author contributions

Gabriella Angiulli, Harveer Singh Dhupar, Investigation, Methodology; Hiroshi Suzuki, Investigation; Irvinder Singh Wason, Conceptualization, Supervision, Funding acquisition, Investigation, Project administration; Franck Duong Van Hoa, Thomas Walz, Conceptualization, Supervision, Investigation, Project administration

## Author ORCIDs

Gabriella Angiulli ⓘ https://orcid.org/0000-0002-0092-9552
Hiroshi Suzuki ⓘ http://orcid.org/0000-0001-5371-6385
Irvinder Singh Wason ⓘ https://orcid.org/0000-0002-3004-857X
Franck Duong Van Hoa ⓘ https://orcid.org/0000-0001-7328-6124
Thomas Walz ⓘ https://orcid.org/0000-0003-2606-2835

## Decision letter and Author response

Decision letter https://doi.org/10.7554/eLife.53530.sa1
Author response https://doi.org/10.7554/eLife.53530.sa2

# Additional files

## Supplementary files

• Transparent reporting form

## Data availability

All data generated or analyzed during this study are included in this Article and the Supplementary Information. The cryo-EM maps have been deposited in the Electron Microscopy Data Bank with accession codes EMD-20950, EMD-20959, EMD-20962. The atomic models have been deposited in the Protein Data Bank under accession codes 6UZH, 6UZ2, 6UZL.

The following datasets were generated:

| Author(s) | Year | Dataset title | Dataset URL | Database and Identifier |
|---|---|---|---|---|
| Angiulli G, Dhupar HS, Suzuki H, Wason IS, Duong F, Walz T | 2020 | MscS: Cryo-EM structure of mechanosensitive channel MscS reconstituted into peptidiscs | https://www.ebi.ac.uk/pdbe/emdb/EMD-20959 | Electron Microscopy Data Bank, EMD-20959 |
| Angiulli G, Dhupar HS, Suzuki H, Wason IS, Duong F, Walz T | 2020 | MscS: Cryo-EM structure of mechanosensitive channel MscS reconstituted into peptidiscs | http://www.rcsb.org/structure/6UZH | RCSB Protein Data Bank, 6UZH |
| Angiulli G, Dhupar HS, Suzuki H, Wason IS, Duong F, Walz T | 2020 | MsbA_Conf1: Cryo-EM structure of nucleotide-free MsbA reconstituted into peptidiscs, conformation 1 | https://www.ebi.ac.uk/pdbe/emdb/EMD-20950 | Electron Microscopy Data Bank, EMD-20950 |
| Angiulli G, Dhupar HS, Suzuki H, Wason IS, Duong F, Walz T | 2020 | MsbA_Conf1: Cryo-EM structure of nucleotide-free MsbA reconstituted into peptidiscs, conformation 1 | http://www.rcsb.org/structure/6UZ2 | RCSB Protein Data Bank, 6UZ2 |
| Angiulli G, Dhupar | 2020 | MsbA_Conf2: Cryo-EM structure of | https://www.ebi.ac.uk/ | Electron Microscopy |

| | | | | |
|---|---|---|---|---|
| HS, Suzuki H, Wason IS, Duong F, Walz T | | nucleotide-free MsbA reconstituted into peptidiscs, conformation 2 | pdbe/emdb/EMD-20962 | Data Bank, EMD-20962 |
| Angiulli G, Dhupar HS, Suzuki H, Wason IS, Duong F, Walz T | 2020 | MsbA_Conf2: Cryo-EM structure of nucleotide-free MsbA reconstituted into peptidiscs, conformation 2 | http://www.rcsb.org/structure/6UZL | RCSB Protein Data Bank, 6UZL |

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
