## [Decision Letter]

**Acceptance summary:**

This paper describes the use of the recently developed peptidiscs for single particle cryo-EM analyses of membrane proteins. The authors solved two membrane protein structures reconstituted in peptidiscs (MsbA and MscS). Cryo-EM structures for both proteins were previously solved using nanodiscs. While the resolution could not be further improved by peptidiscs (they were similar to the resolution obtained in nanodiscs), the cryo-EM maps show density for the peptidiscs. Although the complete/unambiguous assignment of peptidisc helices into the respective densities appeared to be quite challenging, the main finding of the paper is the fact that there is density for the peptidiscs. Hence, peptidiscs differ clearly from nanodiscs and detergent in the sense that there is a (at least partially) ordered and structured interaction between peptide and transmembrane helices.

**Decision letter after peer review:**

Thank you for submitting your article "New approach for membrane protein reconstitution into peptidiscs and basis for their adaptability to different proteins" for consideration by *eLife*. Your article has been reviewed by three peer reviewers, including Volker Dötsch as the Reviewing Editor and Reviewer #1, and the evaluation has been overseen by Richard Aldrich as the Senior Editor. The following individuals involved in review of your submission have agreed to reveal their identity: Markus A Seeger (Reviewer #2); Gerhard Wagner (Reviewer #3).

The reviewers have discussed the reviews with one another and the Reviewing Editor has drafted this decision to help you prepare a revised submission.

Summary:

This paper describes the use of the recently developed peptidiscs for single particle cryo-EM analyses of membrane proteins. The authors developed and describe a novel, gentle peptidisc reconstitution method using a sucrose gradient. Further, they solved two membrane protein structures reconstituted in peptidiscs (MsbA and MscS). Cryo-EM structures for both proteins were previously solved using nanodiscs. While the resolution could not be further improved by peptidiscs (they were similar to the resolution obtained in nanodiscs), the cryo-EM maps show density for the peptidiscs. Although the complete/unambiguous assignment of peptidisc helices into the respective densities appeared to be quite challenging, the main finding of the paper is the fact that there is density for the peptidiscs. Hence, peptidiscs differ clearly from nanodiscs and detergent in the sense that there is a (at least partially) ordered and structured interaction between peptide and transmembrane helices. An additional finding of interest is, that the peptidisc is organized in completely different ways for MsbA and MscS, showing the versatility and adaptability of the peptidisc peptides to keep membrane proteins with different sizes and shapes in solution. The authors nicely describe their findings in the larger context of membrane mimetics (including a comparison to saposins).

Essential revisions:

1) Since there is a cryoEM structure of MscS in a phospholipid nanodiscs, it would be informative to see whether there are differences to their structure may be in a rmsd plot or a color coded surface representation.

2) Did the authors conduct cryo-EM analysis of the respective detergent-purified protein samples? In order to "prove" that the peptidiscs have some advantages over detergent-purified material such a comparison would be useful.

3) Can the authors comment on possible differences of lateral pressure between the peptidisc and nanodisc approach?

4) It would be interesting to see whether peptidiscs increase the stability of embedded membrane proteins. Can the authors show a CD melting curve with a melting temperature, and would the complexes refold after melting?

5) Long NBD distances were detected by DEER using MsbA reconstituted in proteoliposomes (Zou et al., 2009). This should be taken into consideration during the discussion. Shorter NBD distances might be "imposed"/"selected" by the nanodiscs or peptidiscs.

6) The authors describe the density for the peptidisc helices. This is certainly remarkable and very interesting. However, it also raises the question of the impact of the (as it seems to a certain degree variable/partially defined) peptidisc on the entire cryo-EM data processing outcome. Is there any impact expected or observed on particle picking, 2D-class averaging and 3D classification? The authors should comment on this.

7) The pattern of peptidisc-membrane protein interaction appears to be very complex. As the authors point out, the observed helical densities are of different lengths, suggesting that most peptidiscs only interact with a part of their sequence with the protein. Whereas the peptidisc densities for MsbA look rather convincing, they appear quite ambiguous/weak for MscS. Could it be that what the authors claim to be density for the peptidisc is the blurry result of averaging (in particular in MscS with its heptameric symmetry axis)?

8) The method works well with the colored proteins used in this study. The authors propose that the gradient method could be easily used with other, non-colored proteins, if the protein is labeled (via GFP) or the peptide via a fluorescence tag. In particular the last option using a fluorescently labeled peptide would be of high interest to the community as it would show a standard method for the use of peptidiscs with the gradient method. The authors should develop such a peptide that can be used with non-colored proteins and show that fluorescently tagged peptides assemble equally well as the non-tagged peptides.

---

## [Author Response]

Essential revisions:1) Since there is a cryoEM structure of MscS in a phospholipid nanodiscs, it would be informative to see whether there are differences to their structure may be in a rmsd plot or a color coded surface representation.

As suggested, in new Figure 2—figure supplement 7, we now show in panel B) an overlay of our structure of MscS in peptidiscs with a recently published structure of MscS in nanodiscs and in panel C) the RMSDs between the two structures mapped onto our structure. These panels show that the two structures are very similar. In the revised manuscript, we added the following statement: “Our final structure of MscS in peptidiscs is very similar to the published one of MscS in nanodiscs (Figure 2—figure supplement 7B and C)”

2) Did the authors conduct cryo-EM analysis of the respective detergent-purified protein samples? In order to "prove" that the peptidiscs have some advantages over detergent-purified material such a comparison would be useful.

MsbA – We have not performed cryo-EM for MsbA in detergent, because we believe that the evidence described in the manuscript already established that the peptidisc stabilizes MsbA better than detergent. This evidence is:

1) MsbA in peptidiscs has a higher ATPase activity than MsbA solubilized in the mild detergent dodecyl maltoside (DDM). The activity of MsbA in peptidiscs is indeed similar to the ATPase activity of MsbA in the lipid environment provided by nanodiscs (Mi et al., 2017). Since peptidisc-stabilized MsbA is more active than detergent-solubilized MsbA, our conclusion is that peptidiscs better preserve the native structure of MsbA.

2) All averages obtained with peptidisc-stabilized MsbA in negative stain show almost identical conformations, in which the two NBDs are in close proximity. The same conformation is seen for nanodisc-embedded MsbA in negative stain (Mi et al., 2017). In contrast, DDM-solubilized MsbA shows a continuum of conformations with the two NBDs being located at variable distances from each other. Since MsbA in detergent is less active, we believe that this structural heterogeneity represents a destabilization of the MsbA structure by the detergent and thus reflects an artifact. We now explicitly state this in the manuscript: “Even though double electron-electron resonance experiments have found that the NBDs of MsbA reconstituted into proteoliposomes can be distant from each other (Zou et al., 2009), because we only see separated NBDs for detergent-solubilized MsbA, which has a lower ATPase activity, but not for MsbA reconstituted into peptidiscs (Figure 2—figure supplement 2C) or nanodiscs (Mi et al., 2017), we believe that NBD separation is an artifact introduced by the detergent.”

From a technical point of view, the continuous conformational heterogeneity of detergent-solubilized MsbA would make it very challenging to produce a high-resolution cryo-EM structure, and since the structural heterogeneity seems to be an artifact introduced by the detergent, the relevance of such a structure would be questionable.

MscS – Recently, a structure of DDM-solubilized MscS has been published (Reddy et al., 2019). We now show in panel A) of new Figure 2—figure supplement 7 local resolution maps for MscS in peptidiscs and for DDM-solubilized MscS. While the features and the local resolution of the two maps are very similar, the map of MscS in peptidiscs better resolves the connection between transmembrane helices 1 and 2, suggesting that this region is more stable in the context of peptidiscs. In the revised manuscript, we added the following statement: “The resolution of our map is similar to that of the recently published single-particle cryo-EM reconstruction of MscS in DDM (3.4 Å) (Reddy et al., 2019), but our map shows better defined density for the loop connecting transmembrane helices 1 and 2 (Figure 2—figure supplement 7A).”

3) Can the authors comment on possible differences of lateral pressure between the peptidisc and nanodisc approach?

We are not aware of a way to experimentally measure the lateral pressure created by peptidiscs or nanodiscs, so that we can only speculate on this point. We find that both membrane mimetics stabilize the same conformation of MsbA (which is not stabilized by detergent). This result could be interpreted as the two membrane mimetics exerting a similar degree of lateral pressure on the protein. However, the underlying reasons for protein stabilization by the two membrane mimetics could also be different. In a nanodisc, the membrane-scaffold proteins create a physical barrier that encircles the membrane patch, indeed providing a means to generate lateral pressure on the protein, the extent of which would depend on how many lipids are filling in the nanodisc. Peptidisc peptides, on the other hand, simply associate with the transmembrane domain and thus may not exert much lateral pressure on the protein, similar to detergent molecules forming a micelle around the transmembrane domain. However, unlike the flexible acyl chains of detergent molecules, the peptidisc peptides form more rigid amphipathic α-helices. Association of these amphipathic α-helices with the transmembrane domain likely reduces the mobility/flexibility of the transmembrane α-helices, which could be the reason why peptidiscs have a stabilizing effect on transmembrane domains. However, since we are not aware of any experimental approach we could use to test our ideas, we prefer not to include these speculations in the manuscript.

4) It would be interesting to see whether peptidiscs increase the stability of embedded membrane proteins. Can the authors show a CD melting curve with a melting temperature, and would the complexes refold after melting?

In our original publication that introduced peptidiscs as a novel membrane mimetic, we have already recorded melting curves for the reaction center (RC) of *Rhodobacter sphaeroides* (Carlson et al., 2018). Panel C) of Figure 10—figure supplement 1 demonstrates that RC in peptidiscs is indeed more stable than in detergent. Panels D) and E) show that the stabilizing effect of peptidiscs is similar to those of other membrane mimetics, such as SMA, nanodiscs, and proteoliposomes, which all result in a similarly increased melting temperature compared to that of RC in detergent.

The degree of stabilization will vary depending on the detergent as well as on the intrinsic properties and stability of the membrane protein itself. We agree with the reviewers that an in-depth analysis of the stability and refolding of membrane proteins in detergent *versus* different membrane mimetics (with and without lipids) would be interesting. However, such work should be carried out in a systematic manner on several membrane proteins (requiring a high-throughput format), which is beyond the scope of our current Research Advance manuscript.

5) Long NBD distances were detected by DEER using MsbA reconstituted in proteoliposomes (Zou et al., 2009). This should be taken into consideration during the discussion. Shorter NBD distances might be "imposed"/"selected" by the nanodiscs or peptidiscs.

As discussed above and now stated in the manuscript, the longer distances seen for MsbA in detergent correlate with a lower ATPase activity. Also, we cannot think of a reason for why shorter NBD distances would be imposed/selected by nanodiscs and peptidiscs, but would not be imposed/selected by proteoliposomes. Thus, while we cannot rationalize the published DEER results, we believe that the separation of the NBDs observed in detergent-solubilized MsbA is an artifact, since MsbA has a higher ATPase activity in nanodiscs and peptidiscs, in which case we never observe large distances between the two NBDs. To state this more explicitly, we have included the following statement in the revised manuscript: “Even though double electron-electron resonance experiments have found that the NBDs of MsbA reconstituted into proteoliposomes can be distant from each other (Zou et al., 2009), because we only see separated NBDs for detergent-solubilized MsbA, which has a lower ATPase activity, but not for MsbA reconstituted into peptidiscs (Figure 2—figure supplement 2C) or nanodiscs (Mi et al., 2017), we believe that NBD separation is an artifact introduced by the detergent.”

6) The authors describe the density for the peptidisc helices. This is certainly remarkable and very interesting. However, it also raises the question of the impact of the (as it seems to a certain degree variable/partially defined) peptidisc on the entire cryo-EM data processing outcome. Is there any impact expected or observed on particle picking, 2D-class averaging and 3D classification? The authors should comment on this.

In response to the reviewers’ suggestion, we now include a discussion of this issue in the revised manuscript: “The fact that at least some regions of the peptidisc peptides are well ordered provides an additional motivation for using peptidiscs as membrane mimetic. Detergent micelles and nanodiscs are structurally very heterogeneous. Therefore, even though they increase the size of the target membrane protein and thus make the particles easier to see and to pick, they also bury the transmembrane domain in an amorphous density, thus preventing it from contributing signal that could be exploited for alignment, 2D and 3D classification of the particles, especially at early stages of the processing when the resolution of the map is still low. In contrast, while peptidiscs also increase the size of the target membrane protein, the regions of the peptidisc peptides that assume defined positions on the protein also contribute signal that will help in the alignment, 2D and 3D classifications of the particles, similar, for example, to Fab fragments bound to target proteins (Wu et al., 2012). The signal of the ordered peptidisc peptide regions should thus aid in image processing and is therefore an additional advantage of using peptidiscs over detergents or nanodiscs.”

7) The pattern of peptidisc-membrane protein interaction appears to be very complex. As the authors point out, the observed helical densities are of different lengths, suggesting that most peptidiscs only interact with a part of their sequence with the protein. Whereas the peptidisc densities for MsbA look rather convincing, they appear quite ambiguous/weak for MscS. Could it be that what the authors claim to be density for the peptidisc is the blurry result of averaging (in particular in MscS with its heptameric symmetry axis)?

This is a good point and we have indeed thought about this possibility. We had therefore already processed both datasets without imposing 2-fold symmetry for MsbA and 7-fold symmetry for MscS. However, while at lower resolution, the density representing the peptidisc peptides looked very similar and we now show the unsymmetrized density maps in new Figure 2—figure supplement 8. Therefore, the density distribution does not appear to be greatly affected by the symmetrization and thus does represent differences in the arrangement of peptidisc peptides around MsbA and MscS. We added the following statement in the revised manuscript: “To assess whether imposing symmetry during image processing introduced artificial features in the density representing the peptidisc peptides, we also processed the datasets without imposing symmetry. As expected, the resulting maps are noisier and have lower resolution, but the density representing the peptidisc peptides is very similar to that seen in the symmetrized maps (Figure 2—figure supplement 8).”

8) The method works well with the colored proteins used in this study. The authors propose that the gradient method could be easily used with other, non-colored proteins, if the protein is labeled (via GFP) or the peptide via a fluorescence tag. In particular the last option using a fluorescently labeled peptide would be of high interest to the community as it would show a standard method for the use of peptidiscs with the gradient method. The authors should develop such a peptide that can be used with non-colored proteins and show that fluorescently tagged peptides assemble equally well as the non-tagged peptides.

We have indeed already begun work on this approach and have generated a fluorescently labeled peptidisc peptide. We used it to reconstitute MsbA and observe bands on the gradient that represent free peptide and MsbA in peptidisc. We would like to test additional fluorescently labeled peptides and test those with additional protein targets, but we now mention our preliminary experiments in the Discussion and present the results as Figure 1—figure supplement 2. In the revised manuscript, we added the following statement: “Although more work will be needed to fully establish this approach, initial attempts look very promising (Figure 1—figure supplement 2).”